# Portable and Non-Intrusive Fill-State Detection for Liquid-Freight Containers Based on Vibration Signals

**DOI:** 10.3390/s22207901

**Published:** 2022-10-17

**Authors:** Yanjue Song, Ernest Van Hoecke, Nilesh Madhu

**Affiliations:** IDLab, Ghent University—imec, 9000 Gent, Belgium

**Keywords:** fill-state detection, non-intrusive measuring, impulse response, level prediction

## Abstract

Remote, automated querying of fill-states of liquid-freight containers can significantly boost the operational efficiency of rail- and storage-yards. Most existing methods for fill-state detection are intrusive, or require sophisticated instrumentation and specific testing conditions, making them unsuitable here, due to the noisy and changeable surroundings and restricted access to the interior. We present a non-intrusive system that exploits the influence of the fill-state on the container’s response to an external excitation. Using a solenoid and accelerometer mounted on the exterior wall of the container, to generate pulsed excitation and to measure the container response, the fill-state can be detected. The decision can be either a *binary* (empty/non-empty) label or a (quantised) prediction of the liquid level. We also investigate the choice of the signal features for the detection/classification, and the placement of the sensor and actuator. Experiments conducted in real settings validate the algorithms and the prototypes. Results show that the placement of the sensor and actuator along the base of the container is the best in terms of detection accuracy. In terms of signal features, linear predictive cepstral coefficients possess sufficient discriminative information. The prediction accuracy is 100% for binary classification and exceeds 80% for quantised level prediction.

## 1. Introduction

Rail containers and tank-cars are widely used for liquid freight transport and storage. Upon their arrival by railway, the containers are usually detached from the locomotives and wait for the following operation, while the locomotives are immediately driven away to perform other work. In large factories and rail-yards, this gap between container arrival and its processing makes it difficult for the workers to keep track of which containers need emptying and which are ready for re-use. Currently, the problem is usually addressed by a manual check, which can be extremely inefficient when a large number of emptied and filled containers are mixed together in the yard. It also takes long to empty a container due to its huge capacity. This makes it challenging, especially for high-viscosity content, to tell whether the container has been fully emptied. In such cases manual checking can be rather subjective, leading to errors in fill-state estimation. All this results in decreased operational efficiency. Therefore, a remote and automatic fill-state detection system is of great interest to the relevant industries. In addition to a high detection accuracy, a low complexity in installation and maintenance is of great importance in this scenario. Primarily, the focus is on empty/full detection. Estimating the level is of added value; however, here, a generous tolerance in the level resolution is afforded. Instead of an *exact* prediction of the level, it would suffice, for the purposes of the freight industry, to *roughly* quantify the level of the content according to discrete (quantised) *fill-states*.

There have been many studies on liquid level detection. Some popular methods using optical-fiber liquid level sensors [1], optoelectronic level detection [2], impedance change [3], and a few sound reflection-based methods [4,5,6] require direct access to the liquid. Techniques based on Helmholtz resonance [7] put strong restrictions on the shape of the container (presence of an open, constricted neck). Requiring such direct access to contents or introducing openings in the container are difficult to meet in industrial practice, and even dangerous for flammable or toxic contents. This limits the application scenarios of these systems.

In contrast, acoustic methods can provide non-intrusive solutions for this problem. The most common acoustics-based fill-state detection set-ups for liquid containers are summarised in Figure 1. The penetrative method in Figure 1a requires to install a pair of acoustic transducers on opposite sides of the outer surface of the container. When an acoustic signal is applied to the container by the transmitter, the wave is propagated through either air (when the liquid level is below the sensor) or liquid (when the liquid level is above the sensor). Since different media have different acoustic attenuation characteristics, change of the transmission medium between the sensors can be detected by analysing the energy loss in the process, which indicates the change of the liquid level. This method is more appropriate for small containers such as those for food products [8]. The larger the container capacity, the more the energy required to produce a suitable excitation, since the signal needs to be strong enough to be detected by the receiver after attenuation along the propagation path. This may limit the portability and applicability of such methods in the freight industry.

Alternatively, a single transceiver may be used, as is mostly exploited in pulse-echo analysis. There are two ways to employ the pulse-echo analysis for liquid level detection. The first one, as shown in Figure 1b, is based on the echo from the metal-air or metal-liquid interface, i.e., the inside of the container wall [9,10]. The decision criterion is based on the transmitted-reflected signal energy ratio. The ratio depends on the material characteristics at both sides of the interface. Therefore, a different energy ratio indicates a different material behind the container wall. The method in [11] had a similar setup but investigated the relationship between the content at the sensor height and the duration of echos.

The other way of pulse-echo analysis is to capture the reflection from the air-liquid interface as shown in Figure 1c. The transducer can be installed at the bottom [11,12] or the top of the container [13]. Given the speed of sound in air or the liquid, the level heights can be calculated from the time-of-flight [12,13]. Commercial products such as [14,15] are already available in the market. To improve the robustness of such methods in noisy conditions, a preprocessing of the signals e.g., by beamforming with sensor array as in [5] or by signal denoising using e.g., Kalman filtering as in [13] can be done. This method also requires the sensor surface to be vertical to the liquid surface, which could be an issue for rail cars, where access to the underside of the container/wagon may not be easy.

The aforementioned methods work best when the content of the container is homogeneous. Thus, if a container were filled with powders, emulsions, etc., the effectiveness of these approaches may be affected. This limits the applicability of such solutions.

Figure 1d shows the other common set-up with two transducers. To reduce the energy attenuation in propagation, the excitation generated by the transmitter only travels a small distance instead of penetrating the whole container. Here, the literature indicates that the Lamb wave mode [16], the resonance frequency [17,18,19] or the spectral peak amplitude [20] is affected by the presence of liquid and may be used to deduce the liquid level. Detection criteria can be deduced from a physical model of the impulse response of the container. The common practice is to first simulate this model and derive a relation between liquid levels and features extracted from the impulse response of the container. To detect the liquid level in practice, a physical impact is applied to the container and the response is recorded. The extracted features are then input to the relation derived from the simulation to obtain a level prediction. For example, ref. [18] estimates the peak frequency shift during liquid loading by finite element method (FEM) for a standing cylindrical container with constraints applied to its base. The approach of [19] adopts a simplified model, the Euler-Bernoulli beam theory, and yields a similar relationship between the peak frequency and the weight of its content when assuming the container is constrained at both ends. However, these physical models are heavily dependent on the geometry and material characteristics of containers, and the calculation could be too complex to transfer established models to unseen ones. It is thus inadequate as a general solution.

An interesting idea is provided in [17] with a similar setup. Instead of computational analysis, the relation between the spectral feature and the liquid level is given by fitting a regression curve from experimental data. This makes the method theoretically feasible for an easy transfer to other containers. However, the approach is only tested on a relatively small container of 8.5 L. Furthermore, the dependence of the above methods on only the resonance peak features (i.e., amplitude or frequency) would require, in general, an excitation of the whole container and (as we demonstrate here) does not generalise to the much larger containers used in rail-freight. However, this basic idea, which has its parallel in the manual check procedure, where a yard inspector would tell if the container was empty or not by knocking on it and listening to the sound is at the root of our study. We aim to similarly identify the effect of the liquid level by an automated capture and analysis of the impulse response. If the prototype is well designed and the signal processing procedures are carefully chosen, such impulse response analysis can provide a portable solution (i.e., a compact and mobile device that can be easily installed onto travelling containers) for fill-state detection.

In this paper, we employ the set-up in Figure 1d, consisting of a transducer applying an impulse excitation using a mechanical impact (‘knock’) at a location on the outer wall of the container (Note that we use the term ‘impulse’ or ‘knock’ interchangeably in the text to describe such an excitation), and a vibration sensor located a little further away, measuring the response. However, unlike [17], we propose to analyse the spectrum of the impulse response, instead of focusing solely on spectral peak amplitude or its shift. The benefit of this approach is that we do not need to excite the whole container; it is still possible to detect a change in the fill-state by analysing the change in the impulse response locally. Thereby we address the following questions (i) which features of the impulse response are best suited for the task of liquid level detection, (ii) where should the data be captured in order to obtain the best discrimination, (iii) to what extent is a (quantised) level detection feasible and how does this depend on the sensor location. For this, we carry out experiments on two different types of containers in realistic settings.

The paper is organised as follows: the system overview is presented in Section 2. Section 3 introduces the excitation generation and response capture paradigm, the feature extraction, and the machine learning models used for the binary decision and quantised level prediction. Section 4 describes the experimental setup and the data capture, and presents a preliminary, illustrative visual analysis on the feasibility of the system. Finally, in Section 5, the system is validated, first on a smaller tank-container and then on a larger tank-car. The learnings are summarised in Section 6 and directions for future work are presented in Section 7.

## 2. System Concept and Overview

As previously stated, we propose to detect the container fill-state by applying mechanical impacts on the outer wall of the container and recording and analysing its response using an appropriate sensor. The system schematic is shown in Figure 2.

The prototype system hardware consists of: an actuator to generate impulses, a receiver to record the vibrations, and a local controller for data acquisition, sensor synchronisation/activation, and necessary data analysis. The components are powered by a battery/powerpack.

The prototype is installed at a fixed location on the container. Generally speaking, there are two factors to consider when choosing the location: (i) the measured impulse responses at this location should be sensitive to liquid level change, and (ii) the location is easily accessible for convenient installation and system maintenance. While the lower part of the tank is evidently more favourable in consideration of (ii), we consider also a location midway up the container to contrast the difference in performance at different locations.

The test protocol is as follows: when queried, a series of *N* mechanical impacts, uniformly spaced in time, is applied to the exterior wall of the inspected container by the actuator. The receiver starts recording simultaneously, capturing the vibrations caused by the knocks. The recordings are then analysed to predict the fill-state.

Note that the impulse response characteristics would vary from one container *model* to the other. The impulse response also depends on the material properties such as density, bulk modulus, shear modulus, and viscosity. Thus, before deployment, a calibration experiment is required for each concerned container model, with its typical content. This calibrated model can then be stored for later look-up during the analysis.

We note that in a practical, large-scale deployment, one such system would be permanently attached to each container, and would be remotely queried—as needed—by a central server/base station in the yard. The necessary communication between each such system and the base station can be achieved by, e.g., mobile data (3G–5G), wireless LAN, etc. With such necessary infrastructure on-site, a user can remotely query the status of multiple containers in the yard in a short time. The choice of what processing needs to be *local* and what processing needs to be on the central server is a design decision that depends on the available bandwidth for data transfer, the energy trade-off acceptable and the relative costs of the local and central units.

### Prototype Hardware Details

We choose to use a solenoid [21] as the actuator and an acclerometer [22] as the receiver. The solenoid was chosen because it could generate a mechanical impact that could be regarded as wide-band excitation. The solenoid and the accelerometer should be installed at a close distance so that a small amount of energy can still generate a clear signal at the accelerometer. The distance between the actuator and the receiver is set to 20 cm. They are both fixed on the curved surface by custom-designed and 3D-printed fixtures.

To emphasise the low computational effort required, a Raspberry Pi 4 [23] is used as the controller and for recording the impulse responses. For the purposes of this proof-of-concept study, the data were downloaded onto a personal computer, where the analyses were carried out. In principle, however, given a trained model, a controller like the Raspberry Pi is computationally capable of the analyses. The whole system is powered by a power bank, resulting in a portable device that can be permanently installed on the container.

## 3. Signal Analysis and Level Prediction

### 3.1. Signal Pre-Processing

Prior to the analysis stage, the raw recording is first pre-processed. This consists of applying a second-order Butterworth high-pass filter with cut-off frequency at 50 Hz to remove any DC offset and low-frequency noise. Zero-phase filtering was performed to remove non-linear phase effects typical to IIR filters. The recordings are then split into individual time-series segments, centred around each impulse instant. Since impulse application and signal recording begin simultaneously, the recording always starts with an impulse. Given the close distance between the actuator and the receiver, the time of impact can be approximately located at multiples of the impulse intervals. Thus, by setting the theoretical impact time at the centre of a segment, a recording $x={\left(x(0)\;,\ldots \;,x(NP-1)\right)}^{T}$, where *N* is the number of impulses and P is the number of samples between two impulse instants, can be segmented into N−1 segments. An example recording that contains N=20 impulses is shown in Figure 3a. This is a recording sliced into 19 segments as detailed above. Figure 3b shows an example of one such segment, where the impulse is located approximately at the middle of the segment. Whereas the interval between two impulses is a design factor (one that will be analysed later), the segment length is fixed to 0.16 s for all recordings. For a sampling frequency of 3200 Hz, this corresponds to K=512 samples.

Before further analysis, a von Hann window (of length *K* samples) is applied to each segment to ameliorate edge effects. Features are then extracted for each segment and are input to the calibrated model, which outputs the likelihood of each class, for that segment.

### 3.2. Feature Extraction

Previous studies have focused on changes in the spectral peak amplitude [20] or the spectrum peak position (frequency) [17] to detect change in the liquid level. However, as can be seen in Figure 4, which depicts the Bartlett power spectra [24] obtained by the system for a freight container at different fill levels (see Section 4 for the details of the recording setup), *no* dominant spectral peak can be observed. Thus, the above methods, based solely on detecting changes in the frequency or amplitude of the spectral peak, would fail. We hypothesise, instead, that the spectral *envelope* of the impulse response, which captures the spectral evolution of the impulse response over the entire frequency range (up to the Nyquist frequency) would be a more robust marker of level indication. A change in level would affect the response in a broadband manner, resulting in a change in the spectral envelope. To obtain a low-dimensional parameterisation of the envelope, two popular spectral features [25], Mel Frequency Cepstral Coefficients (MFCCs) and Linear Prediction Cepstral Coefficients (LPCCs), are investigated.

#### 3.2.1. MFCC

MFCC is a widely used acoustic feature to extract the signal envelope. This representation encodes the spectral envelope on the Mel-scale, which is linearly spaced under 1 kHz and logarithmically above 1 kHz. This scale is derived from the subjective perception of pitch by humans. The corresponding Mel-spectrum can be obtained by filtering the signal through a series of band-pass filters whose boundaries are evenly spaced on Mel scale. Then, the filtered spectrum is transformed into cepstral domain by the discrete cosine transform. The resulting representation is termed the MFCC.

Calculating the MFCC of a segment $x_{i}$ includes the following steps [26,27]. The sliced and windowed frame is transformed into the spectral domain $X_{i}\left(k\right)$ by applying a *K*-point discrete Fourier transform (DFT): (1)$$X_{i}\left(k\right)=\sum_{n=0}^{K-1}x_{i}\left(n\right)e^{\frac{-j2\pi nk}{K}}$$
where *k* denotes the discrete frequency-bin index. The relationship between physical frequency *f* and Mel frequency $f_{mel}$ can be approximated as
(2)$$f_{mel}=2595\log_{10}\left(1+\frac{f}{700}\right)\;.$$

Thereby, the Mel-spectrum S(m) is obtained by passing the magnitude spectrum through the Mel-filter bank that contains *M* filters to warp the physical frequency into the Mel frequency according to this non-linearity. Finally, the discrete cosine transform of the log Mel-spectrum, computed as: (3)$$c_{\mathrm{MFCC}}\left(q\right)=\sum_{m=0}^{M-1}\log_{10}\left(S\left(m\right)\right)\cos\left(\frac{\pi q(m-0.5)}{M}\right)$$
yields the Mel-frequency cepstral coefficients. In this work, we choose M=21 and q∈(1,⋯,M−1). As $c_{\mathrm{MFCC}}(0)$ is essentially a measure of the level of the signal, it carries no envelope-related information and is therefore not considered. Thereby we obtain scale-invariant features for the envelopes.

#### 3.2.2. LPCC

Within the audio processing and analysis community, the tendency is to use the MFCCs as the typical acoustic features for most classification tasks, with the argument being that these are derived from human perception and, therefore, are biologically ‘optimal’. However, it is *not necessary* that features optimised for human hearing would be similarly optimal for machine audition. Therefore, another common acoustic feature, the LPCC, is also taken into consideration. LPCC also captures the spectral envelopes, but is derived from linear predictive analysis of the signal. Linear prediction analysis imposes an auto-regressive (AR) model on the sequences, whereby each signal sample is predicted as the linear combination of the preceding samples. For a frame $x_{i}\left(n\right)$, an AR model of order *L* generates an estimate of $x_{i}\left(n\right)$ by a linear combination of previous *L* samples: (4)$${\hat{x}}_{i}\left(n\right)=\sum_{\ell =1}^{L}a_{\ell ,i}x_{i}\left(n-\ell \right)\;.$$

The linear predictive coefficients $a_{\ell ,i}$ can be efficiently estimated for each frame $x_{i}$ by the Levinson-Durbin recursion [28]. LPCCs are the cepstral representation of the AR model coefficients, obtained in a recursive fashion as [26]: (5)$$\begin{array}{cc} c_{0} & =\log_{e}L\\ c_{p} & =a_{p}+\sum_{i=1}^{p-1}\frac{i}{p}c_{i}a_{p-i},\;\mathrm{for}\;1<p\leq L\\ c_{p} & =\sum_{i=m-p}^{m-1}\frac{i}{p}c_{i}a_{p-i},\;\mathrm{for}\;p>L\;. \end{array}$$

We adopt L=20 and p∈(0,1,…,L), to keep the same dimensionality as the MFCC features. Note that the coefficient $c_{0}$ provides no information on the signal envelope. Hence, the *L*-dimensional vector $c_{\mathrm{LPCC}}={\left(c_{1},c_{2},\ldots ,c_{L}\right)}^{T}$ is selected as the LPCC feature vector.

### 3.3. Level Detection

We propose to tackle ‘empty/full’ detection as well as the prediction of the liquid level as *classification* problems. The choice of a binary classifier model for the ‘empty/full’ detection is obvious. Treating the level prediction as a classification is less intuitive and deserving of an explanation.

We note that if a prediction of liquid level on a continuous scale was desired, a *regression* model would be more appropriate. However, such a model also significantly increases the requirements on the quality and the number of the corresponding calibration experiments, which is infeasible from a practical point of view. Moreover, the feedback we obtained from the freight industry managers indicated that a quantised level detection is already sufficient for their purpose of improving productivity. Given level prediction as a secondary goal and the available tolerance in its precision, posing this as a classification problem is, therefore, more efficient.

### 3.4. Classifier

The logistic regression (LR) model with L2 penalty [29] is adopted as the classifier in our system. The output of the LR model can be interpreted as the likelihood of each class.

The training targets are generated from ground-truth data labels according to the goal of each model. For the binary classifier, a threshold is first determined. Then, the discrete liquid level labels are classified as ‘non-empty’ if the level is higher than the threshold, or ‘empty’ otherwise. Anticipating our findings in Section 4.3, the height of the sensor is chosen as the threshold for the binary classification.

For liquid level prediction, the discrete data labels are directly regarded as prediction classes. Each liquid level (e.g., 60%, 40%, 10%, and 0%, as used in our case) is taken as an independent class. This labelling scheme corresponds to a multiclass problem. We adopted multinomial Logistic Regression [29] which utilises the cross-entropy loss to realise a direct prediction of the likelihood of each class.

Given multiple segments in one recording, we can increase the robustness of the decision by *average-pooling* the likelihoods of each class across the segments. The class with the highest average likelihood is taken as the fused decision (i.e., this class represents the container state) for the recording.

## 4. Data Gathering and Preliminary Analysis

### 4.1. Setup and Ground Truth

All the experiments were performed on the site of GCA Netherlands (Groupe Charles André: https://gcanederland.nl/ (accessed on 15 December 2021)) in Moerdijk with the assistance of GCA Netherlands B.V. and Ovinto. First, we conducted a preliminary test experiment on a small container as shown in Figure 5 to investigate the feasibility of the proposed method. Its capacity is 30 m^3^, and the content was water. The prototype was installed at the bottom of the container, and its impulse response was measured when the container was full, half-full, and empty. Due to the good thermal conductivity of steel and the heat capacity difference between water and air, the temperature of the tank wall has an evident difference above and below the water surface. The water level can thus be detected from the exterior by measuring the tank wall temperature with an electronic thermometer. Using the height of the tank as the reference, the height of water levels are normalised into percentage. This measurement provided the ground truth labels for the experiment.

Next, we validated the proposed method on a full-size rail tank wagon (shown in Figure 6). The tank is made of carbon steel, and its capacity is around 93 m^3^. The tank was filled with water (to 60% of its height) at the start of the experiment and successively emptied to simulate different fill-states.

The effect of the location of the sensor was one of the questions we investigated. Therefore, we tested two locations in the lower part of the tank, as shown in the schematic Figure 7. Prototype 1 was installed at the bottom of the tank, and prototype 2 at 35% of the height of the tank. Other than their locations, the two prototypes were identical in design. The two installed prototypes also showed good consistency during an offline (lab conditions) sensor calibration test, wherein the two prototypes measured the impulse response of the same structure from the same place. This means that any difference in detection performance between the two prototypes can be attributed to the install location.

The ground truth is obtained similarly as for the container. As part of the test protocol, the water was emptied in a step-wise manner. The outlet was opened for 20 min each time, corresponding to fill-states of 60%, 40%, then 10% of the tank height, and lastly until the tank was totally empty. There was always a five-minute waiting time between the closure of the tank outlet and the impulse response measurement. This allows the water to settle from the turbulence caused by rapid drainage and avoids its interference in the impulse response measurement. After the water had become calm, several sets of impulse responses were recorded by our prototypes. The two prototypes measured the impulse responses alternately, thus avoiding any overlaps between the recordings. The influence of sensor coupling variations was also simulated in the experiments. The solenoids and the actuators were uninstalled and then re-attached to the same location after every three recordings being taken, which guaranteed at least two independent installations for each level.

### 4.2. Recording Protocol

For each test recording, the solenoid generates a sequence of *N* knocks at fixed intervals of T seconds. The accelerometer, consequently, records the vibration signal for N·T seconds. Sampling rate was 3.2 kHz, and N=20 for all recordings.

The spacing between consecutive knocks should be chosen to avoid overlap between the impulse responses. This indicates a conservative (large) value for the interval. However, the longer the interval, the longer it would take for a test run, reducing the measuring efficiency. Since the duration of the container response was not known *a priori*, we experimented with different intervals. For reasons of efficiency, we considered two impulse intervals (0.8 s and 1 s) for the preliminary test. For the experiment on the tank-car, a 1-s interval and a shorter interval (0.4 s), were tested. Analysis indicated that the recorded signals show no measurable overlap of the impulse response for all these cases.

### 4.3. Feature Comparison

We reiterate the key research questions we shall investigate and validate experimentally: (i) which set of features is most discriminatory for our task? (ii) what is the effect of the installation location on the detection accuracy? (iii) how can we increase the robustness of the detection (i.e., what is the minimal number of impulses required for a reliable decision)?

We start with a visual analysis to provide an intuitive comparison of the two acoustic features and the effect of the installation location by data visualisation. For this purpose, Principal Component Analysis (PCA) [30], a dimensionality reduction technique, is employed. PCA is able to transform high-dimensional features into a reduced feature space where the variances of new features are arranged in the descending order. Thereby, the first few components describe the major trends in the distribution of the original data, but in a low dimensional space. In other words it enables us to project high-dimensional data into a hyper plane describing the maximal variance directions. An analysis of the correlation matrix of the features indicates that the eigenvalues of the first two components account for over 85% of the observed variance. Consequently, we choose here to project the acoustic features onto the first two principal component axes and thereby visualise the distribution in a 2D space.

Both feature representations of the measured impulse responses on the tank are extracted and visualised in Figure 8 and Figure 9. Each point in the plot represents the features for one impulse response segment. All data are presented in the figures. We can already make the following observations:

#### 4.3.1. Effect of Sensor Location

A good sensor location means that there is, first of all, a clear discrimination between the response of an empty tank and that of a non-empty one when measuring at the same location and, ideally, a finer discrimination can be found to distinguish between different liquid levels. We observe in Figure 8 and Figure 9 that there is a clear connection between the sensor installation height and the clustering boundary in terms of liquid level: for prototype 1, installed at the bottom of the tank, there is a relatively clear distinction between the data collected on the empty tank and on the non-empty tank, for both features; prototype 2 was installed at 35% of the height of the tank, and the data points of the impulse responses of the empty tank are largely overlapped with the impulse responses of the tank at 10% liquid level. But there is a clear difference between the data points of responses measured with liquid level at 40% and at 10%. Therefore, we can conclude that the impulse response of the container will change significantly when the relative position of the liquid level and the sensor changes, i.e., when the liquid level drops from above the sensor to underneath, or vice versa.

#### 4.3.2. MFCC vs. LPCC

Given the relationship between the impulse responses and the relative sensor location, we took the sensor installation height as the decision boundary. The data points of LPCCs (Figure 9) seem more tightly clustered and linearly separable as compared to MFCCs (Figure 8). Certainly, some information is discarded in the dimension reduction process, which can be crucial to the decision-making. Nevertheless, this figure may be seen as an indication of LPCCs being the better feature for fill-state detection. Since the features are extracted from the same set of data, LPCCs show higher robustness (more concentrated feature space of each cluster) whereas the data representations in MFCC seem widely separated by factors other than the liquid level.

## 5. Evaluation

Following the illustrative analysis, we now present a thorough evaluation of the system. The prime goal is to detect the binary (empty/non-empty) state of the container. Predicting the quantised liquid level is useful (supplementary) information.

The proposed system is first validated on data gathered from the small container. Next, the data gathered on the larger tank-car is evaluated. The effect of the different sensor locations, previously depicted graphically, is now quantified. Furthermore, the minimal number of impulses required for a robust classification will be investigated. This parameter should be optimised to guarantee a good balance between a robust decision and the low energy consumption.

Fused decision was described in Section 3.4 as a means to improve the robustness of the system, by aggregating likelihoods over consecutive responses. To investigate the minimum required number of impulses in a practical system, we pool over the requisite number of consecutive impulses to simulate impulse sequences of different lengths in each recording. The fused decision is investigated for all models. For the data gathered from the container, the models are trained on 60% of the recordings with one-second impulse interval, and tested on the rest of the container recordings. For the data gathered from the tank wagon, the data is divided into training and test set by the impulse intervals: 0.4-s-interval recordings are split 50-50 as training/test set, and all one-second-interval recordings as test set to avoid data leakage.

Separate Logistic Regression (LR) models are trained, for each feature set (LPCC, MFCC) and for each task (the binary state detector and the liquid level predictor), respectively. The evaluation details are listed in Table 1 for the data from the container, and in Table 2 for the data from the tank-wagon.

### 5.1. Evaluation on Container Data

The proposed method is first verified by the preliminary test experiment on the small container. The liquid level detection results on the container are depicted in Figure 10 and Figure 11. The classification results are succinctly summarised in the *confusion* matrix, where the x-axis breaks down the data distribution according to their true labels, and the y-axis by their classification results. Thereby, the correct classifications lie on the diagonal of the matrix. The off-diagonal elements indicate false classifications (e.g., in the case of a binary classifier, the off-diagonal elements would correspond to the missed detection and false alarms). Further, the presented values are normalised by the total number of data-points for each prediction label and expressed as a percentage. Consequently the diagonal elements directly indicate the correct detection rates for each label. This representation of the classifier performance by a confusion matrix thus provides a quick overview not only of the accuracy of the system but also *how* the errors are distributed. This is particularly useful for analysing the performance for multi-label classification tasks, such as the quantised level prediction.

#### Binary Detection

Using LPCC as the feature, the proposed system shows higher accuracy on the binary classification (first row of Figure 10) than on the quantised level prediction (second row of Figure 10). It is clear that fused decisions improves the prediction performance for both goals even when only 3 consecutive impulses are taken into consideration. When we take a closer look at the level prediction results, it can be observed that the major prediction error comes from the confusion in the ‘non-empty’ conditions. If we consider the level prediction results with the ‘empty/non-empty’ boundary (the sensor installation height) as the threshold, the binarised results show a similar accuracy to the results from the binary classifier.

For feature comparison, LR models based on MFCCs are also trained and tested in a similar manner. Figure 10 and Figure 11 indicate that LPCCs outperform MFCCs in both tasks (binary classification and quantised level prediction). The difference is more evident for level prediction. Comparing Figure 10e to Figure 11e, the overall accuracy decreases from 80% (LPCCs) to 65% (MFCCs). This result is consistent with the implications of the PCA projections of the two features (Figure 8 and Figure 9).

The preliminary test indicates that the proposed method is feasible for fill-state detection. Fused decisions improves the prediction accuracy for both goals. LPCCs outperform MFCCs as the input feature for the LR classifier. Therefore, LR with LPCC input as the detection model will be employed in the following sections.


Figure 10The confusion matrices of LPCC-based LR models on the data gathered from the container. Results are presented for the binary classification (‘empty/full’) of model 1 in the first row (**a**–**c**) as well as for the quantised level prediction task of model 3 in the second row (**d**–**f**). The first column indicates the performance on the training data, when only a single impulse is considered. The second column indicates the accuracy on the test set, again for a single impulse. The third column shows the benefit of fusion across three impulses, i.e., pooling the likelihoods across multiple impulses before taking the final decision. The result, for the binary classification, is now 100% on the test set and significantly improved on the more challenging level prediction task.
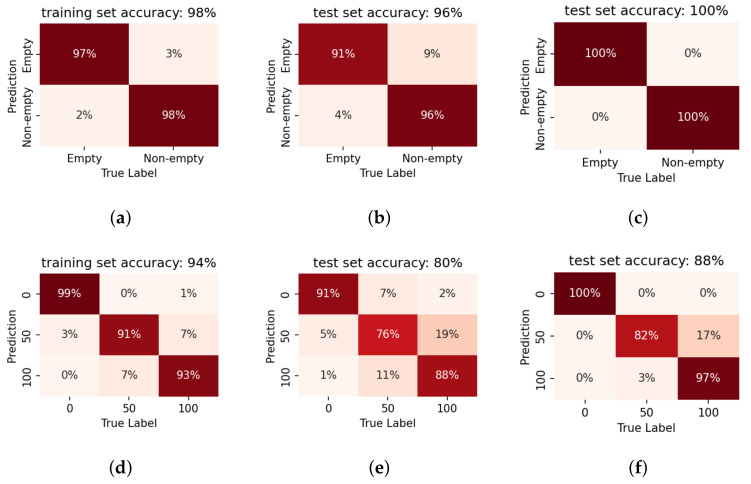

Figure 11The confusion matrices of MFCC-based LR models on the data gathered from the container. The first row (**a**–**c**) shows the the performance of Model 2 in the binary classification task. The second row (**d**–**f**) shows the the performance of Model 4 in the level prediction task. The first, second and third column indicates the performance on the training data, on the test set, and the benefit of fusion across three impulses, respectively. The trends follow that in Figure 10. Whereas MFCC features too, show relatively good performance, this system performs worse compared to the LPCC-based systems. Further, the benefit of fused decisions is evident here as well, especially for the binary classification.
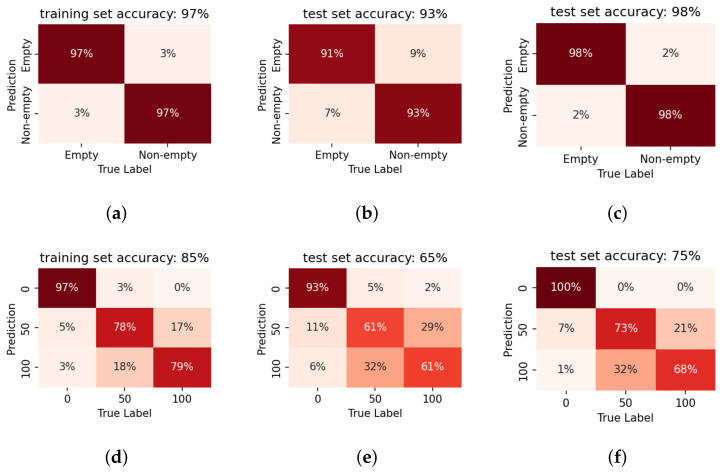



### 5.2. Evaluation on Tank-Car Data

It has already been shown by Figure 9 that the sensor location has an influence on the feature distribution. Therefore, separate models are trained for the two prototypes.

Figure 12 presents the confusion matrices of empty/non-empty binary classification of the models on single impulses. The results are also summarised by sensitivity and specificity in Table 3. Generally speaking, it can be observed that binary classification is a simple task for the proposed system. Nevertheless, prototype 2 that was installed at a higher sensor location performs worse in the binary classification task even though the labels are binarised accordingly. This is in line with the analysis from the PCA visualisation in Section 4.3.

#### 5.2.1. Effect of Fused Decision

Fused decision is introduced to improve accuracy. Fusing likelihoods across different numbers of consecutive single impulse results could simulate a shorter recording and demonstrate the influence of different impulse numbers *N*. The classifiers already show high accuracy on single impulse recordings, so only a marginal increase of three-impulse per recording is simulated as described in Section 3.3 and evaluated. Note that 16 three-impulse recordings can be obtained from one original 20-impulse recording. Figure 12c,f present the fused decision results (only shown for the test set). The fused decision results are also shown in terms of sensitivity and specificity in Table 4, for the test set. As shown in the preliminary test, the binary detection is still an easy task even for the full-size tank-car. A positive effect of fused decisions is also observed with three impulses in each recording.

#### 5.2.2. Sensitivity to Model Mismatch

The sensitivity of the trained *models* to the sensor location is next investigated. Figure 13 shows the binary liquid level detection results when applying the trained prototype 1 model to the data collected by prototype 2. The well-trained classifier fails to predict the data from other sensor locations. The low prediction accuracy indicates that the matching of sensor location and the model is crucial to the correct detection. Therefore, it is essential to ensure that the sensor installation location is fixed for each calibrated system.

#### 5.2.3. Liquid Level Detection

It is also of great interest to see how much extra information we can deduce from the recordings. In this section, we investigate the accuracy of the proposed quantised liquid level estimation system.

Figure 14 shows the liquid level prediction results of the trained LR models by confusion matrices. Since we deal with multi-label classification, we only present the results on the confusion matrix.

This conveys more insights regarding the performance, specifically how the errors are distributed across the various labels. It has already been shown that the lower part of the tank is better for the binary prediction. A similar result can be observed in the liquid level prediction results: prototype 1 installed at the tank bottom performs better than prototype 2 installed at a higher location. Further, the binary ‘empty/non-empty’ classification can be deduced from the level prediction results. Again, if we pool the level prediction results for model 7 using the sensor height as the classification boundary, the binary classification accuracy is in line with the results shown in Figure 12b. This reaffirms the correctness of our choice on the threshold to classify empty/non-empty conditions.

Average-pooling also improves accuracy for liquid level prediction. As shown in Figure 14c, five impulses per recording are sufficient for a reliable liquid level detection for prototype 1. However, the situation is more complex for prototype 2. Comparing Figure 14f with Figure 14g,h, the decision fusion hardly improves prediction accuracy. Especially when the liquid level is at the sensor installation height, the fusion does not bring much additional benefit.

Taking both results into consideration, the optimal sensor location for level prediction is along the base of the container.

## 6. Conclusions

We presented a portable non-intrusive fill-state detection method for liquid containers, to query the container fill-state in a remote and automatic manner. When the container state is queried, a sequence of mechanical impacts (‘knocks’) is applied to the container at evenly-spaced intervals and the container impulse responses are recorded simultaneously. The fill-state information of the container can be deduced from the recorded responses.

Two aims have been set for the system: a main but ‘coarse’ one to detect if the tank is empty, and a supplementary, ‘detailed’ one to detect how much liquid is left in the tank (quantised to a discrete set of fill-levels or fill-states).

Prototypes were designed and built to validate the proposed method, consisting of a solenoid actuator as the excitation source, a piezoelectric sensor as the vibration receiver, a Raspberry PI as an archetypal local controller with limited computational resources, and a powerbank for energy supply. Experiments were performed on a container and a freight tank to verify the method and to optimise the following system configurations: sensor locations, minimal numbers of impulses in one recording for a robust decision and the data representation for different tasks.

We solved both the binary empty/non-empty detection and the quantised liquid level detection as classification problems. The logistic regression model is adopted as the classifier. The visual data analysis by PCA and the more thorough evaluation using trained LR models both show that Linear Prediction Cepstral Coefficients (LPCC) offer a more robust data representation than Mel Frequency Cepstral Coefficients (MFCC) as features. The experiments show that the models are most sensitive at the decision boundary, which is at the same height as the sensor. Therefore, the best sensor location is the bottom of the container given our prime goal of binary fill-state classification. Furthermore, decision fusion could improve detection accuracy for both the binary classification and the liquid level detection. Using the best sensor location (the bottom of the container) and feature (LPCCs), the proposed system yields a 100% accuracy on the binary prediction task if fusing three impulses, while fusing 10 impulses can achieve an accuracy of 80% on the quantised liquid level prediction task.

We note that the proposed method offers a degree of built-in robustness to the external interference endemic in the industrial environments where the system will be deployed. Firstly, extraneous impulses captured in a recording can be detected and filtered out using the synchronisation between the impact generator and the data capture. Secondly, as shown in the evaluation, fusing likelihoods across multiple impulses further improves the robustness of the detection. The high accuracy of the test results, where the experiments were performed in an *active rail-yard* and during *normal working hours*, offer some proof of this robustness.

## 7. Future Work

The proposed system can be an important contributor to improving rail-intelligence and our paper is a first step in this direction. Through proof-of-concept experiments with prototype hardware we have chiefly demonstrated the *feasibility* of the concept. The results indicate that a reliable fill-state detection can be obtained in real-life working conditions, given a proper calibration for a container-content pair. In the context of facilitating a large-scale deployment we expect the calibrations done on one container-content pair to hold for similar container-content pairs, as long as the same model of container is used. This limits the calibration to only a few container-content pairs, which can be acquired over a period of time. Since the purpose of a container (the type of freight it carries) very rarely varies, once a container is calibrated we can expect the parameters to hold for a relatively long period of time. Another way to do the calibration is to have an automation (at least for the binary detection) that makes recordings/measurements whenever the container is emptied and filled; this can ensure an up-to-date calibration for that container.

As more data is gathered, we can establish more general relationships between the acoustic features and liquid level for different combinations of container model and content pairs, which can allow for a generalisation of the calibrations. A mobile set-up is also feasible, in which case the system can be designed as a handheld device, allowing for desired containers to be checked by a field operative. As mentioned above, with sufficient data, the operative could e.g., select models from the model database to use the correct parameters, or these could be derived from the more generalised relations if essential parameters of the inspected container (capacity, insulation layer information, general geometry shape, etc.) are provided. Thus, we hope for interesting developments in this field, in collaboration with the industry.

## Figures and Tables

**Figure 1 sensors-22-07901-f001:**
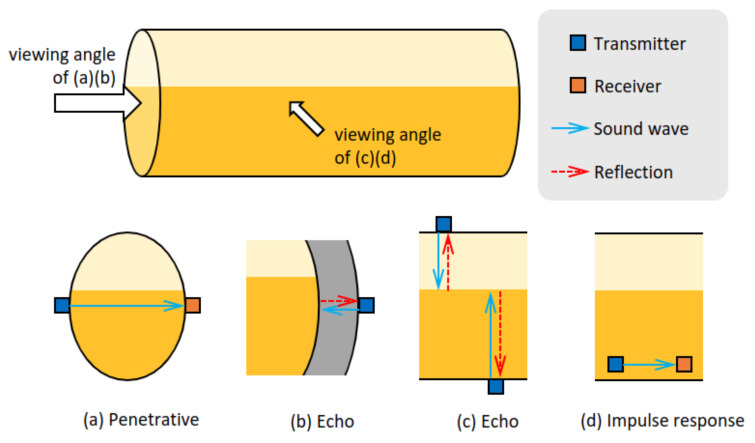
Acoustic-based liquid container fill-state detection set-ups.

**Figure 2 sensors-22-07901-f002:**
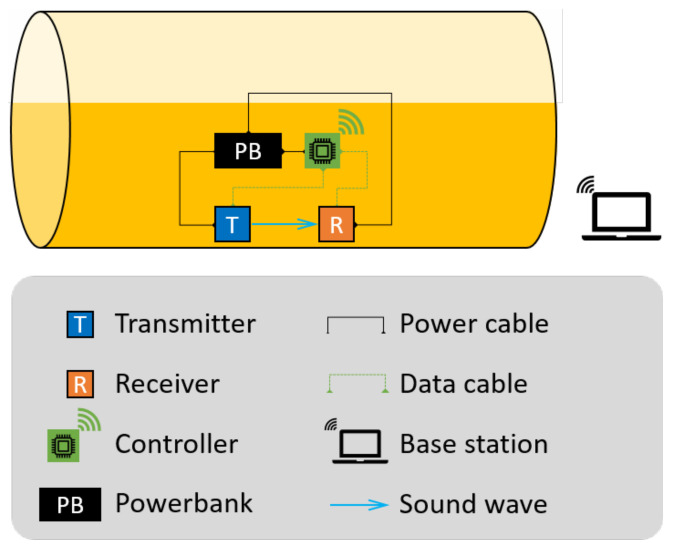
Schematic of the proposed fill-state detection system.

**Figure 3 sensors-22-07901-f003:**
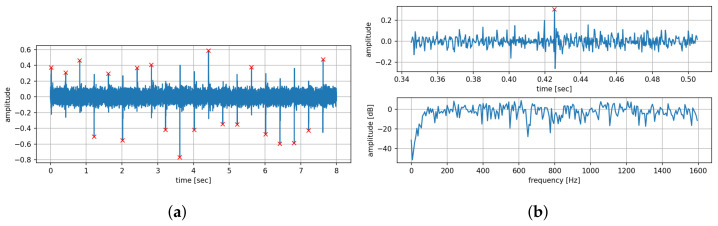
An example of the captured impulse response, where the peaks are marked by red crosses. (**a**) Captured signal for a sequence of N=20 impulses. (**b**) One impulse in the recording in time domain (top panel) and spectral domain (bottom panel). Note how the recording starts with an impulse. For our algorithm, we segment the complete recording such that an impulse lies approximately at the centre of a segment, leading to N−1 segments in general.

**Figure 4 sensors-22-07901-f004:**
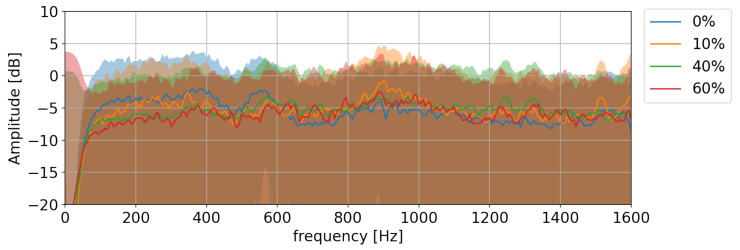
Bartlett power spectra (and 95% confidence intervals of individual periodograms) of the impulse responses recorded at different liquid levels on a tank-car (Section 4). The lines indicate the Bartlett power spectrum and the coloured shaded regions the confidence intervals of the individual periodograms. Note the absence of a dominant spectral peak in the power spectra, indicating that methods based on peak frequency shift or peak amplitude will fail.

**Figure 5 sensors-22-07901-f005:**
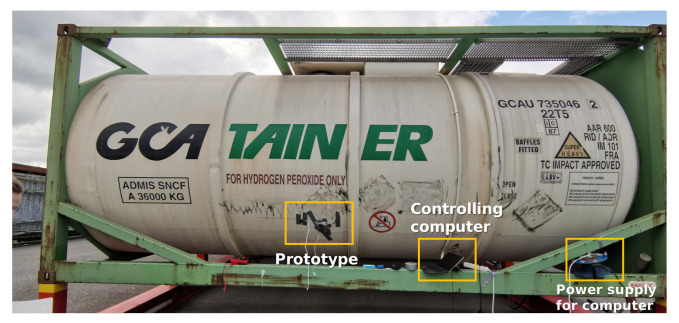
The container for the preliminary test.

**Figure 6 sensors-22-07901-f006:**
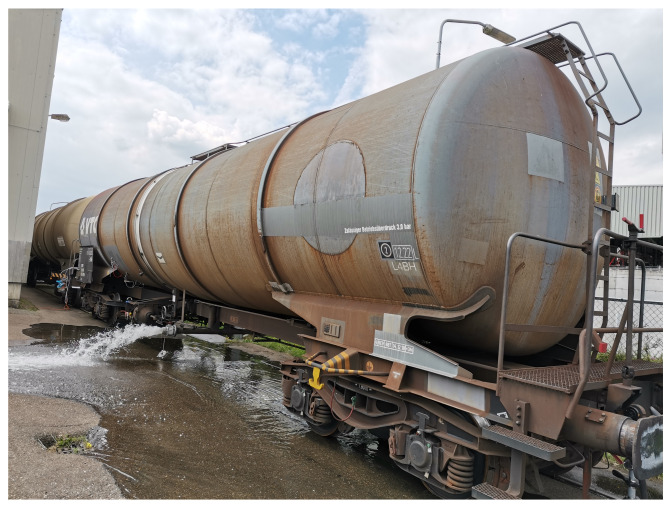
The tank-car used for the experiments.

**Figure 7 sensors-22-07901-f007:**
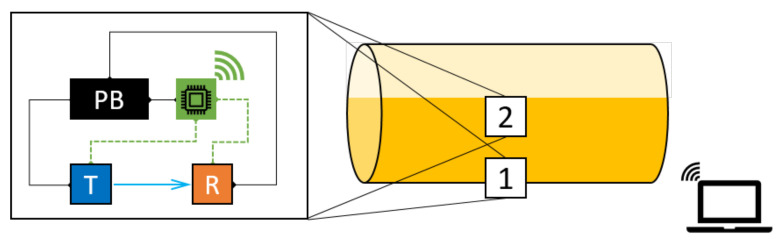
Schematic of the prototype installation on the tank. The two tested installation points are indicated by number 1 and 2. Position 2 is at 35% of the container height.

**Figure 8 sensors-22-07901-f008:**
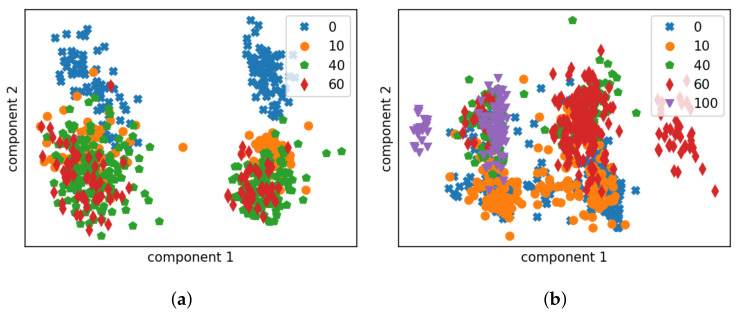
The MFCC distribution of data gathered on the tank wagon, presented in 2D after PCA projection. Each marker corresponds to data-points for a particular residual level of liquid, measured as a percentage of the height of the tank. (**a**) Data collected by prototype 1 installed at the bottom. (**b**) Data collected by prototype 2 installed at 35% height.

**Figure 9 sensors-22-07901-f009:**
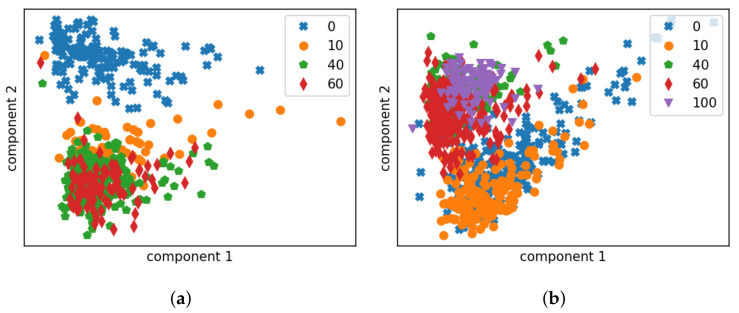
The LPCC distribution of data gathered on the tank wagon, shown in 2D following PCA projection. Each marker corresponds to data-points for a particular residual level of liquid, measured as a percentage of the height of the tank. (**a**) Data collected by prototype 1 installed at the bottom. (**b**) Data collected by prototype 2 installed at 35% height.

**Figure 12 sensors-22-07901-f012:**
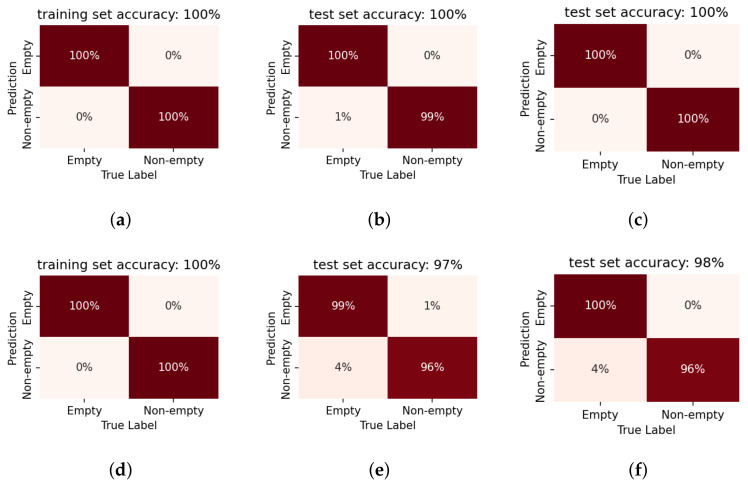
The confusion matrices of LPCC-based LR models on the data gathered from the tank wagon. Results are presented for the binary classification task, for **prototype 1** (first row) by model 5 and **prototype 2** (second row) by model 6. The first column indicates the performance on the training data, when only a single impulse is considered. The second column indicates the accuracy on the test set, again for a single impulse. The third column shows the benefit of fusion across three impulses. The trends are the same as in the case of the container experiments, with high accuracy after decision fusion. **100% accuracy** is obtained by Prototype 1, installed at the bottom of the container. MFCC features yield poorer results—consistent with prior observations on container data—and are, therefore, not presented.

**Figure 13 sensors-22-07901-f013:**
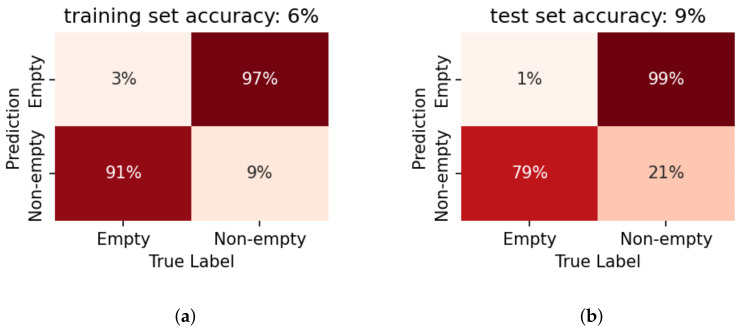
Effect of model mismatch. Binary classification results on recordings collected by prototype 2 when using the classifier model trained for prototype 1 (model 5). (**a**) performance on the training set. (**b**) performance on the test set. The low prediction accuracy indicates the system is sensitive to the install location.

**Figure 14 sensors-22-07901-f014:**
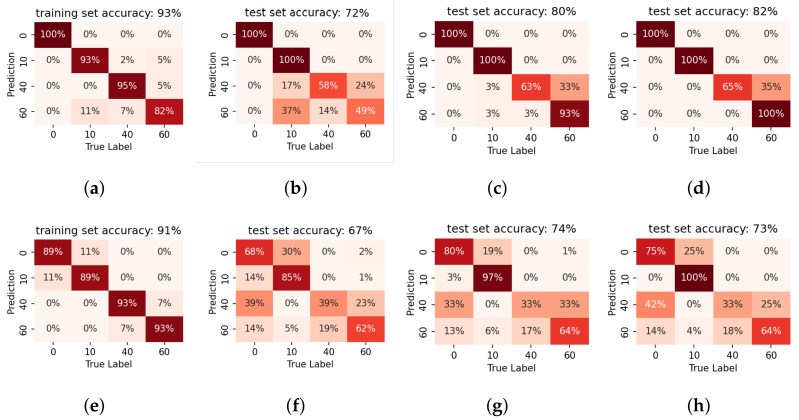
The confusion matrices of LPCC-based LR models on the data gathered from the tank wagon for the level prediction test. Results for prototype 1 by model 7 are presented on the first row (**a**–**d**) and those for prototype 2 by model 8 on the second row (**e**–**h**). From left to right, each column illustrates the performance on the training set, the test set, and the fused decision across 5 impulses and 10 impulses. Prototype 1 shows a relatively high accuracy for level prediction, with some confusion at the 40% mark. Fused decisions is beneficial here as well, with a larger number of impulses increasing the accuracy still further. Prototype 2 has varying results with evident confusion around the decision boundary, again highlighting the dependence of the results with the sensor install location.

**Table 1 sensors-22-07901-t001:** Evaluation details (container). Columns 4 & 5 indicate the *number* of training and test impulses.

Model	Prediction Goal	Feature	# Training Impulses	# Test Impulses
1	binary	$c_{\mathrm{LPCC}}$	228	275
2	binary	$c_{\mathrm{MFCC}}$	228	275
3	level	$c_{\mathrm{LPCC}}$	228	275
4	level	$c_{\mathrm{MFCC}}$	228	275

**Table 2 sensors-22-07901-t002:** Evalulation for the tank-car experiments. Only the LPCC features are considered.

Model	Prediction Goal	Feature	# Training Impulses	# Test Impulses
5	binary	prototype 1	228	437
6	binary	prototype 2	228	494
7	level	prototype 1	228	437
8	level	prototype 2	228	494

**Table 3 sensors-22-07901-t003:** Sensitivity and specificity of Models 5 and 6.

Model	Set	Sensitivity	Specificity
5	training	100.00	100.00
5	test	97.94	100.00
6	training	100.00	100.00
6	test	87.76	98.01

**Table 4 sensors-22-07901-t004:** Sensitivity and specificity of Models 5 and 6, fused decision using three impulses per decision.

Model	Set	Sensitivity	Specificity
5	test	100.00	100.00
6	test	89.09	98.71

## Data Availability

Not applicable.

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
