# Peer review of "Portable and Non-Intrusive Fill-State Detection for Liquid-Freight Containers Based on Vibration Signals"

_sensors, 2022, doi:10.3390/s22207901_

Round 1

Reviewer 1 Report

Song et al. presented “Portable and Non-Intrusive Fill-State Detection for Liquid-Freight Containers Based On Vibration Signals” In this paper, a portable and non-intrusive device for detecting the fill-state of containers used for liquid freight is proposed. The suggested device is installed on the container's outside wall, and fill-state detection is accomplished by an examination of the container's reaction to a produced impulse excitation. The choice can be binary (i.e. empty/non-empty labels) or a (quantified) forecast of liquid level heights. The suggested solution is said to have the benefit of not needing the container to be opened or otherwise requiring access to its inside, making it suitable for large-scale deployment and maintenance.

According to the author, the algorithm and designed prototype were validated in real-world proof-of-concept trials. The trials demonstrated that the system accuracy was reliant on the location of the sensor placement. Installation along the container's bottom was discovered to be the optimal option in terms of detection accuracy and ease of installation and maintenance. Overall, it has been observed that the prediction accuracy at the ideal sensor site is 100% for binary classification and reaches 80% for quantised level prediction. The manuscript is well-written, and the results support the claims of the authors.  The method presented here carry important potential especially for quantized liquid level estimation. I have the following minor comments.

*Text size should be adjusted in most of the figures for better readability. E.g. Figure 6, legends of Figure 4.

*Details of the solenoid that was used as the actuator and an accelerometer should be given.

Reviewer 2 Report

General:

-        The Abstract section is too long (346 words), try to reduce it and make it more synthetic, limiting it to maximum 200 words, as required by the Instructions for the authors.

-        How do you cope with containers that have a protective layer (i.e., mineral thermal insulation), and/or the layers of paint that may extend several millimeters and affect the acoustics of the container. Different types of containers may be shaped in different forms. In other words, how does your solution depend on the constructive features of the container?

-        Figure 2 should be enlarged so that the symbols become more readable. Recommendation: instead of colored rectangles, you may use letters and abbreviations describing the role of the functional blocks, when designing the diagram, to make it more suggestive for the readers.

-        How may the calibration process required in the first time of the installation be modified, or simplified, so as the solution to become more effective for different types of containers and would not require too much customization? This process might be needed for a large-scale implementation of the proposed solution. And in this case, would the approach be suitable for a movable (portable) set-up to be used for individual checking of the filling degree?

-        “The whole system is powered by a power bank, resulting in a portable device that can be permanently installed on the container.” – if the device is portable, then what is the purpose of permanently installing it on the container?

-        The degree of portability of the system is discussible. The user should first determine the best location for the actuator and the sensor, then perform calibration, then measurements. Is this concept correct? If yes, where is the portability? And how can reproducibility be attained, in case of large fleets of containers? Please explain in the body text.

Recommendations:

- Please try to add a small comparison between the advantages and disadvantages of the proposed solution, compared to similar ones.

- Also try to add some information regarding the way the solution may be improved in future, and implemented on a larger scale, by reducing the work for calibration.

- The Conclusion section may also contain recommendations for industry sectors that may make use of your solution.
